

# Assessing alignment-based taxonomic classification of ancient microbial DNA

Raphael Eisenhofer[1,2] and Laura Susan Weyrich[1,2]

[1] Australian Centre for Ancient DNA, University of Adelaide, Adelaide, SA, Australia
[2] Centre of Excellence for Australia Biodiversity and Heritage, University of Adelaide, Adelaide, SA, Australia

## ABSTRACT

The field of palaeomicrobiology—the study of ancient microorganisms—is rapidly growing due to recent methodological and technological advancements. It is now possible to obtain vast quantities of DNA data from ancient specimens in a high-throughput manner and use this information to investigate the dynamics and evolution of past microbial communities. However, we still know very little about how the characteristics of ancient DNA influence our ability to accurately assign microbial taxonomies (i.e. identify species) within ancient metagenomic samples. Here, we use both simulated and published metagenomic data sets to investigate how ancient DNA characteristics affect alignment-based taxonomic classification. We find that nucleotide-to-nucleotide, rather than nucleotide-to-protein, alignments are preferable when assigning taxonomies to short DNA fragment lengths routinely identified within ancient specimens (<60 bp). We determine that deamination (a form of ancient DNA damage) and random sequence substitutions corresponding to ~100,000 years of genomic divergence minimally impact alignment-based classification. We also test four different reference databases and find that database choice can significantly bias the results of alignment-based taxonomic classification in ancient metagenomic studies. Finally, we perform a reanalysis of previously published ancient dental calculus data, increasing the number of microbial DNA sequences assigned taxonomically by an average of 64.2-fold and identifying microbial species previously unidentified in the original study. Overall, this study enhances our understanding of how ancient DNA characteristics influence alignment-based taxonomic classification of ancient microorganisms and provides recommendations for future palaeomicrobiological studies.

Corresponding author
Raphael Eisenhofer,
raphael.eisenhoferphilipona@adelaide.edu.au

## INTRODUCTION

Palaeomicrobiology—the study of ancient microorganisms—is a rapidly growing field of research. As with modern microbiology (*Caporaso et al., 2012*; *The Human Microbiome Project Consortium et al., 2012*), palaeomicrobiology has witnessed a renaissance with the development of high-throughput sequencing technology (*Warinner, Speller & Collins, 2014*; *Weyrich, Dobney & Cooper, 2015*). The study of ancient microorganisms has the

potential to shed light on a range of topics, such as the evolution of the human microbiota (*Adler et al., 2013*; *Weyrich et al., 2017*), adaptation and spread of ancient pathogens (*Bos et al., 2011*, *2014*; *Warinner et al., 2014*), the reconstruction of human migrations and interactions (*Dominguez-Bello & Blaser, 2011*; *Maixner et al., 2016*; *Eisenhofer et al., 2017*), and climate change (*Frisia et al., 2017*).

Palaeomicrobiology is especially challenging because ancient DNA is typically fragmented, contains damage-induced substitutions, and is mixed with the DNA of ancient and modern contaminant microorganisms. DNA fragmentation occurs due to the post-mortem cessation of DNA repair, resulting in short fragment lengths that are typically shorter than 100 bp (*Allentoft et al., 2012*; *Dabney, Meyer & Pääbo, 2013*). These short fragments are also subjected to chemical modifications (e.g. deamination), which yield an increased rate of observed cytosine to thymine and guanine to adenine substitutions at the 5′ and 3′ ends of the sequenced DNA molecules, respectively (*Dabney, Meyer & Pääbo, 2013*). Finally, contamination of ancient DNA with modern microbial DNA is a serious issue that must be mitigated with expensive ultra-clean laboratories, rigorous decontamination, and the extensive use of extraction blank and no-template negative controls (*Salter et al., 2014*; *Eisenhofer, Cooper & Weyrich, 2017*; *Llamas et al., 2017*; *Eisenhofer & Weyrich, 2018*; *Eisenhofer et al., 2019*). Collectively, these factors influence the choice of molecular techniques (*Ziesemer et al., 2015*) and bioinformatic tools used for taxonomic classification of ancient microbial DNA (*Weyrich et al., 2017*; *Velsko et al., 2018*).

Identifying the microbial species present within an ancient sample, that is, taxonomic classification, is a standard first step in palaeomicrobiology studies (*Weyrich et al., 2017*). Initially, targeted amplification of the 16S ribosomal RNA encoding gene was used to discover which microbes were present in ancient samples (*Adler et al., 2013*), as is routinely done in modern microbiota studies seeking to characterize microbial communities (*Caporaso et al., 2012*; *Gilbert, Jansson & Knight, 2014*). However, these targeted regions are often longer than the typical fragment length of ancient DNA and can contain polymorphisms that bias the taxonomic reconstruction of ancient metagenomes (*Ziesemer et al., 2015*). Considering these findings, the palaeomicrobiology field has converged on shotgun sequencing as the best-practice approach to reproducibly identify microbial species within ancient samples. While currently more expensive than the targeted PCR approaches, shotgun sequencing also provides genomic and functional information that can be used to reconstruct ancient microbial genomes, observe functional changes through time, and identify non-prokaryotic information within samples (*Warinner et al., 2014*; *Weyrich et al., 2017*).

Methods for analysing shotgun sequencing data broadly fall into three categories: assembly-based, alignment-free, and alignment-based. Assembly-based techniques involve merging overlapping DNA fragments into longer sequences with the goal of assembling whole genomes. Such techniques have been successful in generating new genomes from modern metagenomic samples (*Imelfort et al., 2014*; *Parks et al., 2017*). However, the short, damaged nature of ancient DNA renders assembly-based techniques currently intractable for palaeomicrobiology. Alignment-free methods use features of

the DNA sequences themselves, such as matches of *k*-mers between reference genomes and the DNA sequences from a sample (*Wood & Salzberg, 2014*; *Ounit & Lonardi, 2016*). To our knowledge, there has been minimal testing of alignment-free methods for the taxonomic classification of ancient microbial DNA. In their assessment of taxonomic classifiers for ancient DNA, *Velsko et al. (2018)* tested the alignment-free method CLARK-S and found that while it had no false-negatives on their simulated metagenome, it had the highest number of misclassifications and false-positives. Alignment-based techniques involve the alignment of DNA fragments to previously characterized reference sequences using alignment algorithms, such as Bowtie2 or the Burrows–Wheeler Aligner (*Li & Durbin, 2009*; *Langmead & Salzberg, 2012*), and include MetaPhlAn (*Truong et al., 2015*), MG-RAST (*Meyer et al., 2008*), DIAMOND (*Buchfink, Xie & Huson, 2015*), and MALT (MEGAN alignment tool) (*Vågene et al., 2018*). A recent study benchmarked these alignment based tools and found that MALT performed better for short, fragmented DNA (*Weyrich et al., 2017*). MALT is an alignment-based tool that allows researchers to query DNA sequences against reference databases using a method similar to Basic Local Alignment Search Tool (BLAST) (*Altschul et al., 1990*), albeit >100 times faster (*Vågene et al., 2018*). MALT can either align nucleotide sequences to nucleotide databases (MALTn) or nucleotide to amino acid databases by translating the DNA prior to alignments (MALTx). A potential advantage to using amino acid alignments for palaeomicrobiology is the greater sequence conservation of peptides due to codon redundancy. This property may help smooth over small changes occurring in DNA sequence over time, allowing ancient sequences to be more easily aligned to modern references. However, the already short nature of ancient DNA yields even shorter amino acid sequences (e.g. 60 bp DNA translated = 20 amino acid sequence), which may not provide a sufficiently high alignment score for taxonomic classification (*Huson et al., 2007*; *Pearson, 2013*). Additionally, DNA damage can result in alignment errors, further lowering alignment scores. To date, there has been no formal testing of nucleotide vs. amino acid alignments for taxonomically classifying short sequences typical of ancient DNA.

Here, we test how characteristics of ancient DNA influence alignment-based taxonomic classification using both simulated and published ancient DNA data sets. We demonstrate that the MALTn (nucleotide-to-nucleotide alignment) approach can improve taxonomic identifications over MALTx (nucleotide-to-protein). We also corroborate previous findings that deamination minimally impacts alignment-based taxonomic classification and that reference database choice is an important consideration when attempting to reconstruct ancient microbial communities (*Warinner et al., 2017*; *Velsko et al., 2018*). Finally, we perform an extensive reanalysis of previously published shotgun DNA sequences from ancient dental calculus with these factors in mind.

## METHODS

### Simulated and published metagenomes

We downloaded 6,897 complete bacterial genomes from the NCBI Assembly (17th May 2017). A total of 29 oral and environmental genomes were used as input for

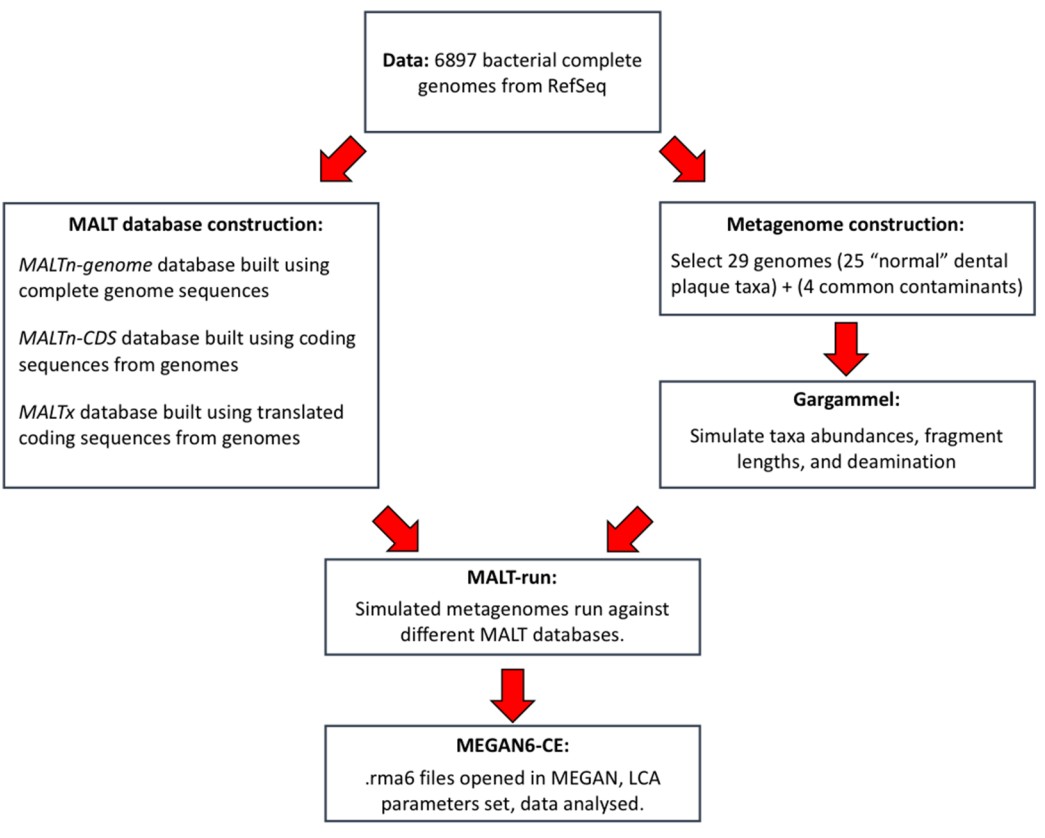

**Figure 1 General overview of simulated data construction and analysis.**

Gargammel (*Renaud et al., 2017*) to generate simulated ancient metagenomes of 1.5 million fragmented sequences each. Briefly, selected bacterial genomic sequences were assigned abundances representative of a typical dental plaque community (Table S1) and then fragmented into metagenomes containing either strict 30, 50, 70, 90 bp (base pair) fragments, or an empirical ancient DNA fragment length distribution that mimicked commonly observed ancient DNA fragmentation (–loc 4, –scale 0.3 in Gargammel) (Fig. S1; Fig. 1) (*Renaud et al., 2017*). To benchmark the influence of deamination on taxonomic classification, the simulated metagenomes of different fragment lengths were then deaminated using Gargammel with the following parameters: nick frequency = 0.03, length of overhanging ends (geometric parameter) = 0.25, probability of deamination in double-stranded parts = 0.01, along with three different probabilities of deamination in single-stranded parts: zero for 0% $\delta_s$; 0.1 for light deamination (10% $\delta_s$); and 0.5 for heavy deamination (50% $\delta_s$) (*Briggs et al., 2007*). Additionally, a real mapDamage profile from the LaBrana sample (*Renaud et al., 2017*) was simulated using Gargammel for the 'empirical' deamination (~20% $\delta_s$). Overall, this resulted in a total of 20 different simulated metagenomes: (five different fragment lengths, 30, 50, 70, 90, and empirical) multiplied by four different deamination magnitudes (0%, 10%, 20%, and 50% $\delta_s$) = 20 (Metagenome 1–20; Table S2).

Simulated metagenomes and the genomes used to build the metagenomes are available via Figshare: https://doi.org/10.25909/5b84c9c196f54, https://doi.org/10.4225/55/5b0caf73b7247, https://doi.org/10.4225/55/5b0ca9b2cd6dc. The collapsed (merged) DNA sequences for 22 published ancient dental calculus samples were downloaded from Online Ancient Genome Repository (https://www.oagr.org.au/experiment/view/65/) (*Weyrich et al., 2017*). Two ancient dental calculus samples from *Warinner et al. (2014)* were also downloaded from the SRA (SRR957739 and SRR957743).

## Reference databases

For the analysis of simulated metagenomes, we created databases that contained the exact same bacterial genomes present in the 20 simulated data sets. We downloaded 6,897 complete bacterial genomes from the NCBI Assembly (17th May 2017), along with their coding sequences (CDS) and translated CDS. These three sources of sequences were used to construct different MALT databases: MALTn-genome (complete genomes); MALTn-CDS (nucleotide coding sequencing from these genomes); and MALTx (translated CDS from these genomes).

For the analysis of previously published dental calculus data, we used sequences from the four following databases: (1) 2014nr (NCBI non redundant protein BLAST database, downloaded 11th November 2014; (*Weyrich et al., 2017*)); 2017nt (NCBI nucleotide BLAST database, downloaded 6th June 2017); (3) HOMD (all human oral microbial genomes (1,362) from the Human Oral Microbiome Database, downloaded July 2017); and (4) RefSeqGCS (47,713 Complete-, Chromosome-, and Scaffold-level assemblies downloaded from NCBI RefSeq database (366 archaeal; 47,347 bacterial)). Genome accessions used for the RefSeqGCS and HOMD databases are available from Figshare (https://doi.org/10.25909/5b84ddf58ac49, https://doi.org/10.25909/5b84d19aaff2a).

## Generation of divergent sequences

Nucleotide substitution rates are known to differ between different species of bacteria, making accurate modelling of bacterial genome evolution is a difficult task. Here, we apply a simplified approach that ignores insertions and deletions, and instead creates a worst-case scenario for benchmarking the effects of nucleotide substitutions on taxonomic classification. We chose a rate of $10^{-7}$ substitutions per site per year, representing the mean of known evolutionary rates for bacterial genomes (*Duchêne et al., 2016*). We assumed an average bacterial genome size of three million bp, thus $10^{-7} \times 3,000,000 = 0.3$ substitutions per genome per year. Scaling for multiple years yielded the following number of substitutions introduced per genome: 10,000 years = 3,000 substitutions (0.1% of genome); 30,000 substitutions (1% of genome); and 300,000 substitutions (10% of genome). We used these numbers to randomly mutate (substitutions only) the bacterial genomes using EMBOSS msbar (*Rice, Longden & Bleasby, 2000*). These 'mutated' genomes were then used as input for Gargammel, fragmented to the empirical ancient DNA fragment length distribution (Fig. S1), and deaminated using the heavy deamination magnitude (50% $\delta_s$) (Metagenome 21–23, Table S2).

## Data analysis

MALT-build v 0.3.8 was used on the reference sequences mentioned above with the default parameters. MALT-run v 0.3.8 was used to align the simulated and real data against the different databases using default settings and outputting BLAST text files (-a). The resulting BLAST text files were converted to RMA6 files using the MEGAN tool blast2rma, as this allows least common ancestor (LCA) parameter adjustment across multiple files. All RMA6 files were then imported and analysed in MEGAN6 CE V6.8.13 (*Huson et al., 2016*). We used the weighted LCA algorithm (80% LCA percentage: -alg weighted -lcp 80) (*Huson et al., 2016*); the minimum support percent filter was set to 0.1% (-supp 0.1) for the published ancient dataset to remove poorly supported assignments (i.e. taxonomic assignments require at least 0.1% of a percent of the total sequences to be considered), and 0.01% for the simulated metagenomes; the minimum expected value (*E*-value) was set to 0.01 (−*e* 0.01); and all other values were left at default. Analysis of the simulated data found that a minimum support percent of 0.1% removed false positive taxonomic assignments for nucleotide-to-nucleotide alignments (Fig. S15), justifying this threshold for the reanalysis of the previous published data. Little research has been done regarding the effect of LCA parameters on taxonomic classification, and such research deserves its own study. Regardless, the parameters chosen for this study represent a conservative approach implemented to reduce noise within the data set.

For the UPGMA tree comparison, species found in extraction blank controls (Table S9), but not environmental controls, were removed (filtered) from the ancient dental calculus samples (*Weyrich et al., 2017*). This filtering approach can be conservative and does not eliminate issues of cross-contamination between samples and controls occurs (*Eisenhofer et al., 2019*). However, the lack of oral taxa classified in the extraction controls makes it unlikely to have affected the downstream analyses. (Table S9). The UPGMA tree was then constructed by exporting the Bray–Curtis distance matrices constructed at the species level from MEGAN6 into SplitsTree4 (*Huson & Bryant, 2006*). The divergences between predicted and simulated abundances were calculated using log-odds scores: log odds = $\log_2$ (predicted abundance/simulated abundance) and the Pearson correlation coefficient.

## RESULTS

### MALTn classifies shorter DNA sequences than MALTx

We assessed the alignment performance of nucleotide-to-nucleotide (MALTn) and nucleotide-to-protein (MALTx) alignments using simulated metagenomes that mimic the characteristics of ancient DNA (Fig. S1). When comparing the differences between nucleotide or protein alignments on the empirical fragment length distribution simulated metagenome, MALTn-CDS (CDS only) classified 5.55-fold more total sequences than MALTx (protein translation of CDS only) (Fig. 2). We investigated this phenomenon further by assessing nucleotide and protein alignments using simulated metagenomes with strict fragment lengths (30, 50, 70, and 90 bp). MALTx analysis was unable to align sequences from the 30 to 50 bp simulated metagenomes and only aligned 33% of sequences from the 70 bp simulated metagenome (Table 1). In contrast, MALTn-CDS aligned 86% of sequences at 30 bp (Table 1). As nucleotide alignments additionally provide the
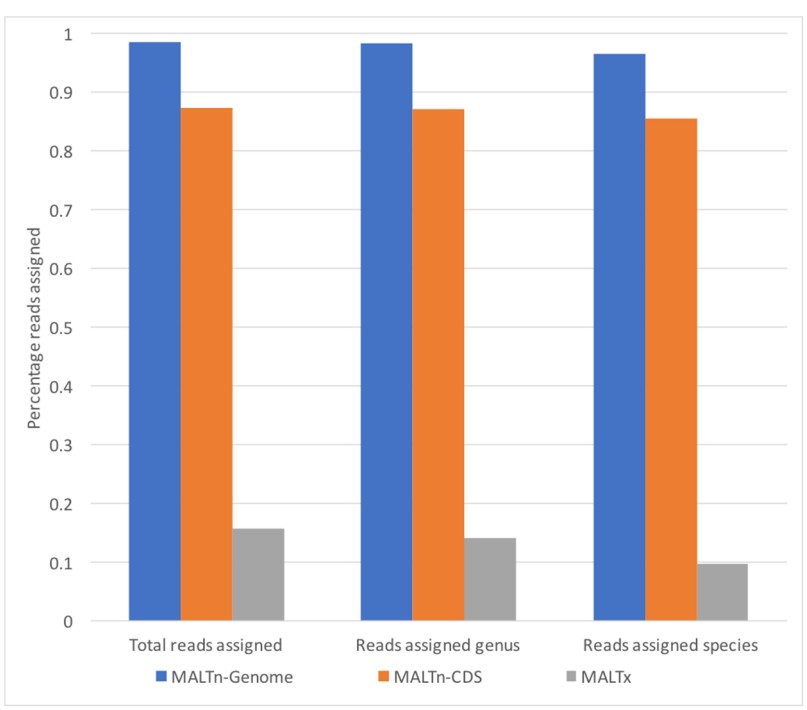

**Figure 2 Percentage of reads assigned taxonomy using simulated metagenomes of empirical ancient DNA fragment length against different MALT databases.**

**Table 1 Percentages of total reads assigned at different taxonomic levels with different read length cut-offs.**

| Fragment length | Reads assigned total | Reads assigned genus | Reads assigned species |
|---|---|---|---|
| 30 bp_MALTn-Genome | 100 | 100 | 97 |
| 30 bp_MALTn-CDS | 86 | 86 | 83 |
| 30 bp_MALTx | 0 | 0 | 0 |
| 50 bp_MALTn-Genome | 100 | 100 | 98 |
| 50 bp_MALTn-CDS | 88 | 88 | 86 |
| 50 bp_MALTx | 0 | 0 | 0 |
| 70 bp_MALTn-Genome | 100 | 100 | 98 |
| 70 bp_MALTn-CDS | 90 | 90 | 88 |
| 70 bp_MALTx | 33 | 31 | 25 |
| 90 bp_MALTn-Genome | 100 | 100 | 98 |
| 90 bp_MALTn-CDS | 91 | 91 | 89 |
| 90 bp_MALTx | 82 | 75 | 55 |
| Empirical_MALTn-Genome | 99 | 98 | 97 |
| Empirical_MALTn-CDS | 87 | 87 | 86 |
| Empirical_MALTx | 16 | 14 | 10 |

additional opportunity to identify non-coding sequences, we also compared nucleotide alignments to full genomes, rather than CDS. Nucleotide alignments including non-coding sequences (MALTn-genome) were able to classify 6.19-fold more total sequences

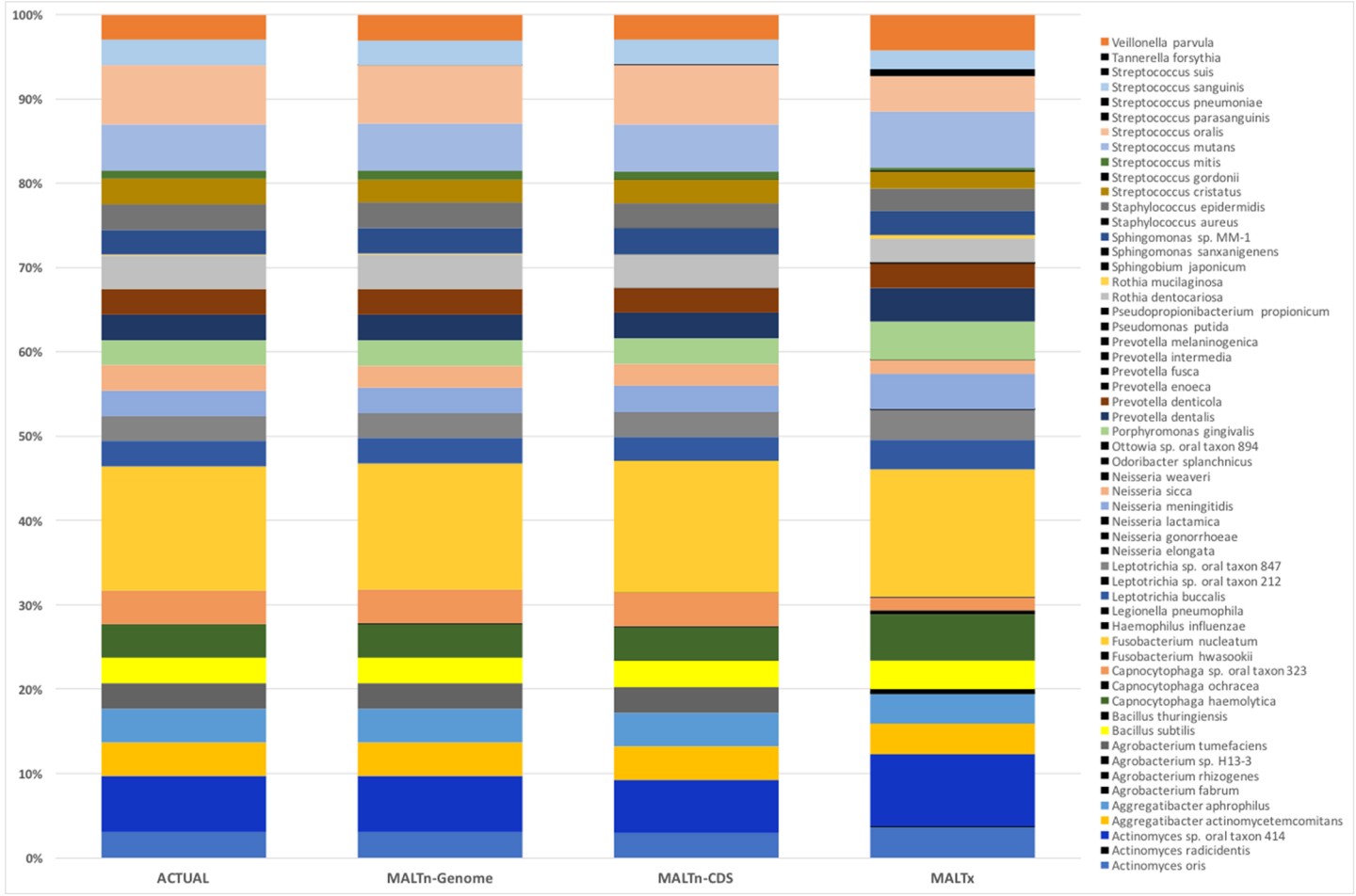

**Figure 3 Species level taxonomic classification of empirical fragment length simulated metagenome.** Species coloured black were not used as input for constructing the simulated metagenomes and are misclassifications.

than MALTx for the empirical fragment length distribution (sevenfold and 9.7-fold more sequences at the genus and species level, respectively) (Fig. 2; Table 1).

## MALTn taxonomic classifications are more accurate than MALTx

While MALTn can classify more sequences than MALTx, the accuracy of these assignments has not yet been examined. We tested the accuracy of these assignments by comparing them to the 'ground truth' (i.e. the actual composition of the simulated metagenomes). Overall, MALTn more accurately reconstructed the simulated, empirical length metagenome composition than MALTx (0.998; Pearson correlation; −0.48 sum of log-odds scores between MALTn-CDS and actual metagenome) (Fig. 3). Even though sequences below 50 bp were not classified, MALTx was able to faithfully reconstruct the simulated metagenome, albeit with poorer abundance predictions compared to nucleotide classifications (0.943; Pearson correlation and −6.66 sum of log-odds scores between MALTx and actual metagenome) (Fig. 3). MALTx misclassified more sequences, resulting in 24 taxa being falsely predicted, whereas only 11 taxa were misclassified using nucleotide

**Table 2 Effects of deamination on taxonomic classification of empirical ancient DNA read-length distribution.**

| Fragment length | Reads assigned total (%) | Reads assigned genus (%) | Reads assigned species (%) |
|---|---|---|---|
| MALTn-genome_0δs | 98.6 | 98.4 | 96.6 |
| MALTn-genome_10δs | 98.4 | 98.2 | 96.5 |
| MALTn-genome_20δs | 98.5 | 98.3 | 96.5 |
| MALTn-genome_50δs | 97.7 | 97.5 | 95.7 |
| MALTn-CDS_0δs | 87.4 | 87.1 | 85.5 |
| MALTn-CDS_10δs | 87.2 | 86.9 | 85.3 |
| MALTn-CDS_20δs | 87.2 | 86.9 | 85.3 |
| MALTn-CDS_50δs | 86.5 | 86.2 | 84.6 |
| MALTx_0δs | 15.8 | 14.2 | 9.7 |
| MALTx_10δs | 15.2 | 13.7 | 9.4 |
| MALTx_20δs | 15.0 | 13.6 | 9.2 |
| MALTx_50δs | 14.5 | 13.1 | 8.9 |

(MALTn-CDS) (Table S3). By increasing the minimum support percent from 0.01 to 0.1, these false predictions were eliminated for MALTn-genome and MALTn-CDS and reduced to 3 for MALTx (Fig. S15). Additionally, classification accuracy with nucleotide alignments was not impacted by fragment length, as MALTn accurately classified sequences as short as 30 bp (Figs. S2 and S3).

We also tested how non-coding sequences can impact the accuracy of taxonomic identifications. The addition of non-coding sequences to the reference database had a limited effect on the accuracy of taxonomic classifications, as the MALTn-genome classifications were almost identical to MALTn-CDS (0.999; Pearson correlation between MALTn-genome and MALTn-CDS) (Fig. 3); however, fewer misclassifications at the species level were identified using MALTn-genome (11 species for MALTn-CDS vs. two species for MALTn-genome). Overall, these results suggest that MALTn classifications are more accurate than MALTx both in providing fewer misclassifications and by providing more accurate abundance predictions. Additionally, it appears that including non-coding information in reference databases (e.g. MALTn-genome) may also reduce misclassifications.

## Deamination minimally affects alignment-based classification

We next tested the effects of deamination (a commonly observed form of ancient DNA damage) on alignment-based taxonomic classification. We tested three scenarios of deamination: light (10% $\delta_s$), moderate (~20% $\delta_s$), and heavy (50% $\delta_s$) (Table 2). Using the empirical fragment length distribution, heavy deamination did not substantially impact the number of sequences using MALTn (0.9% loss of sequences assigned at the species level for and MALTn-genome; 1.3% for MALTn-CDS; and 9.2% for MALTx) (Table 2). As expected, lower magnitudes of deamination had an even smaller impact (Table 2). We also assessed the impacts of heavy deamination on the assignment of DNA sequences of different lengths. Shorter (30 bp) sequences were more affected for nucleotide
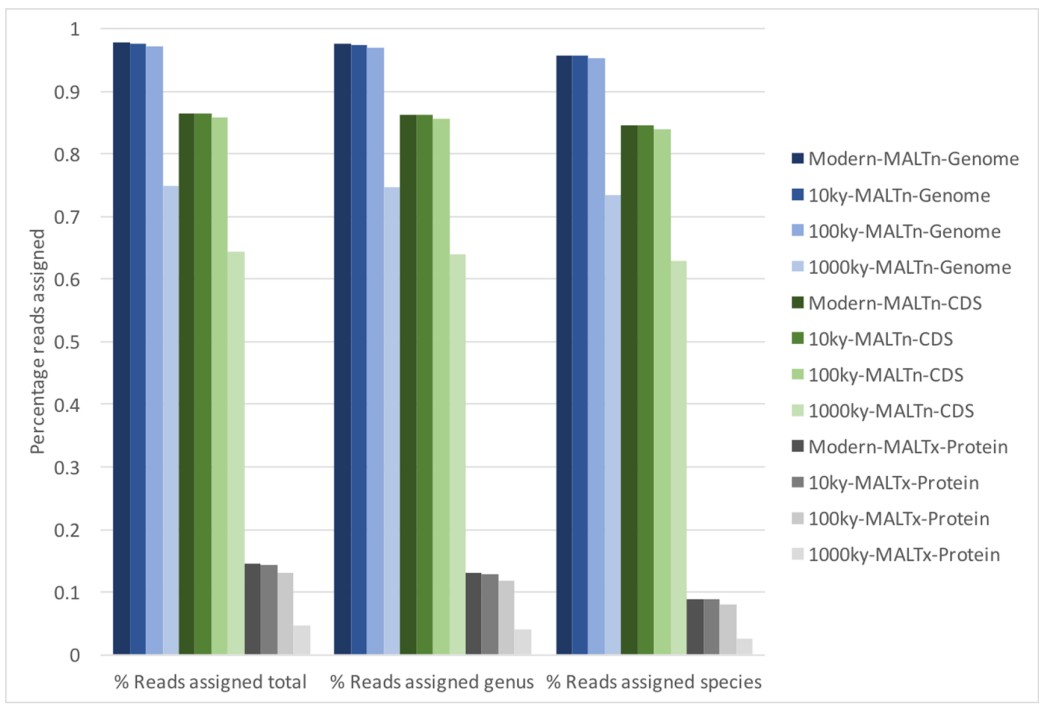

**Figure 4 Percentage of reads assigned taxonomy using divergent and deaminated simulated metagenomes of typical ancient DNA fragment length.**

alignments (9.53% loss of sequences assigned at the species level for MALTn-genome, 8.41% for MALTn-CDS; no alignments for MALTx), but this effect was not observed for sequences longer than 50 bp (Tables S4–S6). Regarding taxonomic composition of the empirical read length metagenomes, heavy deamination did not substantially increase the percentage of misclassifications at the species level (0.06–0.07% for MALTn-genome, 0.29–0.30% for MALTn-CDS, and 2.42–2.48% MALTx). Deamination also did not substantially affect taxonomic composition (Figs. S4–S6). Overall, these results corroborate previous findings that deamination minimally affects alignment-based taxonomic classification (*Velsko et al., 2018*).

## The influence of sequence divergence on taxonomic classification

The effects of sequence divergence on alignment-based taxonomic classification have not yet been explored. To this end, we created divergent simulated metagenomes by introducing random substitution mutations into the same reference genomes used in the above experiments. We chose three different divergence magnitudes: 0.1% sequence divergence (equating to roughly 10ky (1,000 years) of evolution), 1% (100ky), and 10% (1,000ky), and added heavy (50% δ$_s$) deamination, allowing us to examine the worst-case impacts of sequence divergence on taxonomic classification. Overall, MALTn-genome, MALTn-CDS, and MALTx were able to effectively assign taxonomy with minimal loss of alignments (<1% of sequences were unable to be aligned) at 0.1% and 1% sequence divergence (Fig. 4). At 10% divergence (1,000ky), the influence of divergence was more pronounced, as the percentage of sequences not assigned

taxonomically (i.e. sequences with no alignments) increased from 2.28% to 25.1% for MALTn-genome, 13.48% to 35.7% for MALTn-CDS, and 85.45% to 95.4% for MALTx. Even with the loss of sequences assigned with 10% divergence, the taxonomic classifications and abundances remained relatively stable (Figs. S7 and S8), although protein alignments were more affected (0.944 Pearson correlation coefficient between 1,000ky composition and actual simulated metagenome composition for MALTn-genome; 0.944 for MALTn-CDS; and 0.825 for MALTx). As expected, shorter sequences were more affected by sequence divergence and deamination (Fig. S9). Overall, our simulations suggest that random sequence divergence of less than 1% minimally affects alignment-based taxonomic classifications.

## Reference database choice strongly influences taxonomic classification

Because alignment-based methods are highly reliant on reference sequences available in databases, we next sought to test the influence of database choice on taxonomic classification of ancient microbial DNA. To this end, we constructed four different reference databases from different sources: 2014nr, 2017nt, HOMD, and RefSeqGCS. The 2014nr database contains the 2014 non-redundant protein BLAST database, which was used in a recent palaeomicrobiology publication (*Weyrich et al., 2017*) and represents the example of a database used with the MALTx method. The 2017nt databased contains all sequences within the 2017 NCBI nucleotide BLAST database; this is the default for BLAST searches on the NCBI website and does not include chromosome-, scaffold-, or contig-level genome assemblies. The HOMD database contains genomic sequences from the HOMD, which is a curated nucleotide database comprised of oral-associated microbial species and includes all genome assembly levels (complete genomes, chromosomes, scaffolds, and contigs). Lastly, the RefSeqGCS possesses complete, chromosome, and scaffold level genome assembly levels from bacterial and archaeal assemblies within the NCBI RefSeq. The RefSeqGCS database also contains substantially more entries than the HOMD database (e.g. 47,713 vs. 1,362 microbial genomes for HOMD) with a broader diversity of entries (i.e. not primarily oral taxa).

We first tested these different databases on the empirical read length simulated metagenome with and without moderate deamination ($\sim$20% $\delta_s$). The 2014nr performed the worst, with skewed abundances, four false positives, and six false negatives (Fig. S10). In contrast, the 2017nt, HOMD, and RefSeqGCS more accurately recapitulated the simulated metagenome, with the exception of the HOMD, which could not assign reads to *Sphingomonas* sp. MM1 (Fig. S10).

To test the effects of these different databases on the taxonomic classification of real paleomicrobiological data, we aligned the sequences from four published dental calculus samples (three ancient, one modern) (*Weyrich et al., 2017*) against the four databases mentioned above. As expected, the MALTx approach using the 2014nr database assigned the least number of sequences taxonomically, while the MALTn approach using the RefSeqGCS database assigned the most sequences (Fig. 5). In addition, the highest
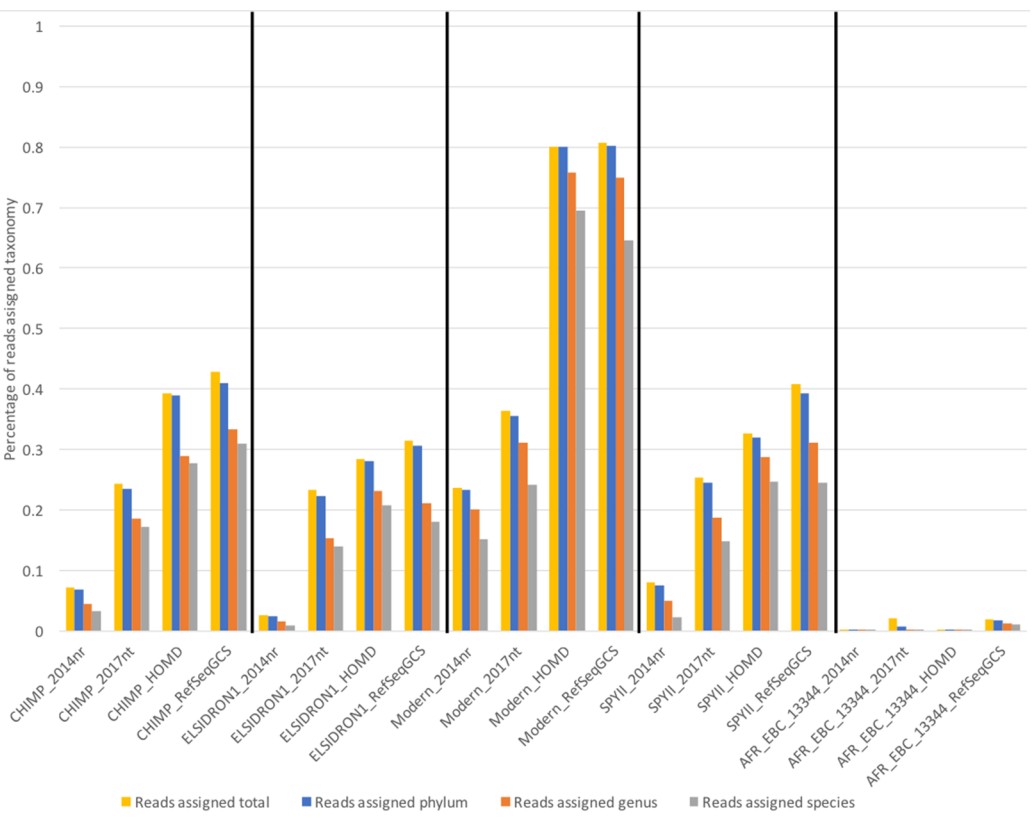

**Figure 5 Percentage of reads assigned taxonomy to different taxonomic ranks for deeply sequenced published data.** Clustered columns represent samples analysed using different reference databases. Colours indicate specificity of assignments.               

percentage of sequences assigned taxonomic classification was observed with the modern sample when using nucleotide alignments with the RefSeqGCS database (80.8% sequences assigned; Fig. 5); this was in stark contrast to average percentage of reads assigned to three ancient oral metagenomes, where on average only 38.3% of sequences were classified. In the ancient samples, the highest number of classified species was observed when ancient sequences were aligned to the HOMD (Table 3), rather than the RefSeqGCS. The higher number of species observed when mapping to the HOMD could be due to either cross-mapping from environmental taxa (as it contains few soil/environmental genomes) or a higher diversity of oral-specific assemblies. Taxonomic compositions in the analysis were also markedly impacted by the database used (Figs. S11–S14; Table S7). Several oral taxa within the HOMD and RefSeqGCS databases are not present within the 2017nt database, such as *Actinomyces dentalis, Bacteriodetes sp. oral taxon 274, Capnocytophaga granulosa, Corynebacterium matruchotii, Methanobrevibacter oralis, Prevotella sp. oral taxon 317,* and *Pseudoramibacter alactolyticus.* This is a likely reason for the 2017nt assigning taxonomy to a smaller percentage of total sequences across all samples (24.3%) when compared to the HOMD (33.4%) and RefSeqGCS (38.3%). Overall, the RefSeqGCS database assigned the most sequences taxonomically and contained the most diverse selection of reference genomes,

**Table 3  Number of genera and species identified in each MALT database.**

**Genus-level**

| Database: | 2014nr | 2017nt | HOMD | RefSeqGCS |
|---|---|---|---|---|
| CHIMP | 46 | 57 | 35 | 52 |
| ELSIDRON1 | 49 | 50 | 42 | 48 |
| MODERN | 23 | 32 | 28 | 29 |
| SPYII | 64 | 64 | 54 | 62 |
| Average | 46 | 51 | 40 | 48 |

**Species-level**

| Database: | 2014nr | 2017nt | HOMD | RefSeqGCS |
|---|---|---|---|---|
| CHIMP | 39 | 59 | 57 | 52 |
| ELSIDRON1 | 42 | 53 | 73 | 69 |
| MODERN | 34 | 58 | 73 | 63 |
| SPYII | 87 | 86 | 74 | 77 |
| Average | 51 | 64 | 69 | 65 |

allowing for more efficient detection of both oral species and potential environmental contaminants. Therefore, we chose the RefSeqGCS for subsequent reanalysis of published dental calculus samples.

## Reanalysis of published dental calculus data with nucleotide alignment

To further test the performance of the RefSeqGCS database, we reanalyzed several published ancient dental calculus samples (total of $n = 24$) (*Weyrich et al., 2017*), including samples from an additional study ($n = 2$) (*Warinner et al., 2014*). We found that MALTn with the RefSeqGCS database substantially increased the number of sequences assigned taxonomically compared to published results (average of 64.2-fold increase with MALTn against the RefSeqGCS vs. MALTx against the 2014nr; Table S8). Despite the increase in sequences assigned using MALTn, the average percentage of unassigned sequences remained relatively high (58.2%), although this was substantially lower than MALTx (94.2%). The MALTn-RefSeqGCS analysis also identified new species in ancient dental calculus specimens, including *A. dentalis, Bacteroidetes sp. oral taxon 274, Capnocytophaga granulosa, Corynebacterium matruchotii, Eikenella corrodens, Lautropia mirabilis, M. oralis,* numerous *Prevotella species, Pseudoramibacter alactolyticus, Slackia exigua,* and *Treponema socranskii.* When a UPGMA tree was constructed using Bray–Curtis distances, ancient agriculturalists were still generally found to cluster independently from hunter-gatherers, with the exception of a single ancient agriculturalist (LBK 1) (Fig. 6). However, the separation between the different types of hunter-gatherers was less pronounced than previously reported (*Weyrich et al., 2017*), and several samples with low oral signals did not fall within either cluster (e.g. Chimpanzee, Spy II, and Afr SF1). Overall, these findings highlight how different analytical strategies can alter the findings of ancient DNA studies and suggest that it will be important to revisit

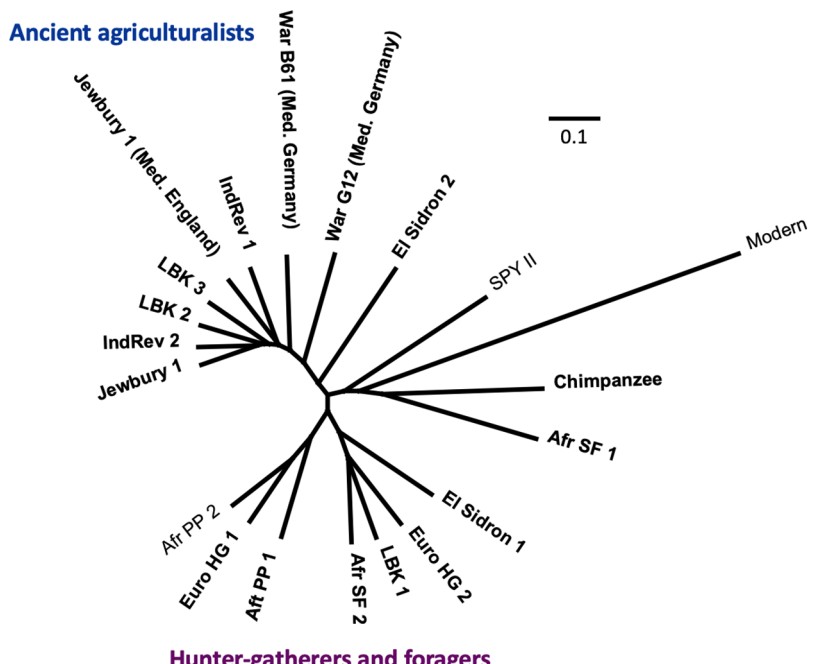

**Figure 6 UPGMA tree of species-level Bray–Curtis dissimilariies calculated from the microbial composition between each sample.** The branch scale bar represents the Bray–Curtis dissimilarity between samples.

previously published datasets as reference databases become larger and analytical techniques are improved.

## DISCUSSION

Using both simulated and real data, this study demonstrated that nucleotide-to-protein alignments currently struggle to assign taxonomy to the short DNA fragments typical of ancient DNA. We found that nucleotide-to-nucleotide alignments using MALTn can faithfully recapitulate simulated metagenomes with high accuracy even when sequences are extremely short (30 bp), contain high levels of deamination, and possess random sequence divergence corresponding to 100,000 years of evolution. We also tested four different reference databases and find that database choice is an important factor to consider for alignment-based taxonomic classification in ancient metagenomic studies; however, we also find that reliable, whole genome information incorporated into database usage drastically improves sequence mappability. Finally, we performed an in-depth reanalysis of a previously published paleomicrobiome study, increasing the number of sequences assigned taxonomically by an average of 64.2-fold and identifying taxa previously unidentified in the original study. We hope that the findings and suggestions provided in this paper will help inform future palaeomicrobiological researchers.

We evaluated the performance of both nucleotide-to-nucleotide and nucleotide-to-protein alignments for taxonomic classification and found that sequences shorter than ~60 bp could not be aligned using a nucleotide-to-protein approach. This can limit

the feasibility of nucleotide-to-protein alignments for some palaeomicrobiological studies given that ancient DNA sequences can be typically shorter than 60 bp. Nucleotide-to-protein alignments are limited by nucleotide translation, shortening the alignment length by a third (e.g. a 60 bp nucleotide sequence = a 20 aa protein sequence) and yielding a lower alignment score (bit-score). Given that the default bit-score threshold for MALT is 50, most short sequences would struggle to obtain a sufficient score to pass filtering. Additionally, amino acid scoring matrices can also influence the final score of the alignment; the default MALTx scoring matrix is BLOSUM62, which optimized for longer sequences (*Pearson, 2013*). The inability to align short sequences may also bias taxonomic composition towards modern environmental and laboratory contaminant taxa, whose sequences are typically longer.

Despite the 5.55-fold loss of sequences assigned using nucleotide-to-protein alignments, the taxonomic classifications were relatively similar to the nucleotide alignments for the simulated data set. However, nucleotide-to-nucleotide alignments lowered the rate of misclassifications. These misclassifications primarily resulted from the lack of non-coding sequences in the protein and CDS nucleotide databases, with misclassifications being supported by sequences that were derived from non-coding genes in the simulated inputs (e.g. tRNA, rRNA, etc.). Recent estimates from 2,671 complete bacterial genomes place the average percentage of non-coding DNA at 12% (*Land et al., 2015*); this represents a non-trivial amount of information that should be harnessed when using reference-based taxonomic alignment. Finally, we also demonstrated nucleotide-to-nucleotide alignments using MALT can faithfully recapitulate simulated taxonomic composition using sequences as short as 30 bp, highlighting the applicability of nucleotide-to-nucleotide alignments for ultra-short fragments typical of palaeomicrobiological studies. Pending further optimization to nucleotide-to-protein alignment methods, we currently recommend using a nucleotide-to-nucleotide alignment approach for taxonomic classification of short length ancient DNA and the inclusion of non-coding information in reference databases to reduce potential misclassification and to increase the amount of information used in alignments.

In this study, we tested the impacts of deamination on shotgun metagenomic taxonomic classifications. *Velsko et al. (2018)* previously found that deamination (~25% δs) minimally affected taxonomic classification, and our results corroborate their finding using three different deamination rates (10%, 20%, and 50% δs). We demonstrated that even high levels of cytosine deamination (50% δs) did not substantially impact taxonomic classification in longer sequences; however, we observed a loss of ~15% of the species level classifications when analysing 30 bp DNA sequences with this level of deamination. This suggests that the use of uracil-DNA-glycosylase (UDG) (*Briggs et al., 2010*)—an enzyme that cleaves deaminated cytosines to reduce the rate of ancient DNA errors—may not be required for microbial taxonomic classification of ancient remains, as this also reduces the total number of sequences that can be analysed. Additionally, treatment with UDG—either full or partial (*Rohland et al., 2015*)—substantially reduces a key source of ancient DNA authentication, which is critical in palaeomicrobiological studies to verify ancient taxa from modern contamination. The lack of such authentication in

palaeomicrobiological research has already led to contentious claims (*Austin et al., 1997*; *Weyrich, Llamas & Cooper, 2014*; *Eisenhofer, Cooper & Weyrich, 2017*; *Eisenhofer & Weyrich, 2018*). Given the minimal impact of deamination on alignment-based taxonomic classification, and the importance of deamination as a measure of ancient DNA authenticity, we recommend against the use of UDG for future palaeomicrobiological studies that focus on alignment-based classification.

Sequence divergence is another characteristic of ancient DNA that can render taxonomic classification difficult. We tested three substitution-based sequence divergence simulations and found that rates of random sequence divergence corresponding to <100,000 years unlikely to alter palaeomicrobiological classifications. A substantial reduction in the number of identified sequences was observed for samples with sequence divergence simulated at one million years (~20% loss of sequences assigned taxonomically). However, this is at the theoretical limit of DNA preservation (*Allentoft et al., 2012*) and is thus unlikely to hamper most palaeomicrobiological studies. We also found that the shorter sequences were, the more they were affected by sequence divergence and deamination, and this can intuitively be explained by the reduction in raw alignment score due to mismatches to the reference. As such, the use of new molecular techniques to obtain even shorter DNA fragments (e.g. <25 bp (*Glocke & Meyer, 2017*)) may prove especially difficult to classify taxonomically given the combined effects of sequence divergence and deamination. Overall, we found that alignment-based taxonomic classification appears robust against magnitudes of random nucleotide substitution that could be observed in ancient DNA <100,000 years old. Despite this, we did not test the impacts of insertions, deletions, and recombination on taxonomic classifications; all would likely further hinder taxonomic classifications. Future simulations accounting for differences in synonymous/non-synonymous mutations may give amino acid alignments an advantage, given the excess synonymous mutations observed due to purifying selection (*Ochman, 2003*), although amino acid alignment scoring would still have to be optimized to deal with short DNA fragments. Additionally, future studies simulating the effects of insertions, deletions, and recombination on taxonomic classification are warranted.

We found that database choice had a major impact on both the number of sequences that were assigned taxonomically and the taxa classified by MALT. *Velsko et al. (2018)* previously observed biases in databases used between different taxonomic classifiers, and our study sought to test the impact of different databases within a single taxonomic classifier, MALT. The 2017nt BLAST database performed poorly compared to the HOMD and RefSeqGCS, assigning on average 33% fewer sequences taxonomically and lacking numerous key oral taxa. This is likely because the 2017nt BLAST database does not contain draft, unfinished bacterial genomes assemblies, which is a major limitation for ancient dental calculus research given that some important oral taxa currently have only chromosome or scaffold-level assemblies, such as *A. dentalis, Bacteroidetes sp. oral taxon 274, Capnocytophaga granulosa, Corynebacterium matruchotii, E. corrodens, L. mirabilis, M. oralis,* numerous *Prevotella species, Pseudoramibacter alactolyticus, S. exigua,* and *T. socranskii*. While the HOMD database contained substantially fewer reference

sequences compared to the RefSeqGCS (1,362 vs. 47,713, respectively), it performed comparably regarding the number of sequences assigned from ancient dental calculus samples. However, using the HOMD database alone for taxonomic classification of ancient dental calculus can be problematic, as it does not contain many environmental or laboratory contaminant taxa that are typically present in ancient samples, such as *Sphingomonas* sp. MM1, which could not be assigned from the simulated metagenome. These environmental and laboratory contaminant taxa allow for the quantification of contamination and competitive alignment, which can prevent false positive assignments (*Key et al., 2017*). Overall, the larger diversity of the RefSeqGCS database increases its ability to classify the most sequences taxonomically, so we would recommend it over the others tested for future palaeomicrobiological studies. An important caveat to using RefSeq references is that some uncultured organisms can be underrepresented. For example, searching 'Saccharibacteria'—an important oral phylum (formerly TM7)—in the NCBI Assembly yielded 153 GenBank entries, and only two RefSeq entries (October 2018). While greater diversity is typically desirable in a reference database, further work is needed to assess and curate the quality of reference assemblies, especially of scaffold- and contig-level, to ensure reliable and accurate alignment-based taxonomic classification (*Parks et al., 2015*). There is also scope for a concerted effort by palaeomicrobiological researchers to work together in constructing a curated, regularly updated reference database. This could help foster reproducibility and set a standard for future work in the field, similar to what has been accomplished by the HOMD for oral microbiome studies (*Chen et al., 2010*).

We also performed a reanalysis of previously published ancient dental calculus data from (*Weyrich et al., 2017*) to test if our in-silico findings were true for real data, explore the proportion of sequences currently classifiable, and see whether the relationships between samples changed when using the RefSeqGCS database. Nucleotide alignment against the RefSeqGCS database performed considerably better compared to protein alignment against the 2014nr, with an average 64.2-fold increase in the number of sequences assigned taxonomically. As expected, this increase was higher for samples with shorter mean fragment lengths and highlights the importance of using nucleotide-to-nucleotide alignments to more accurately reconstruct ancient samples. Despite the substantial increase in the number of sequences aligned, the average number of sequences that did not have any alignment was still 58.2%. When compared to the latest extension to the human microbiome project where the average number of sequences without alignment was ~25% for 265 supragingival plaque samples (*Lloyd-Price et al., 2017*), this suggests that substantial reference bias exists for ancient calculus samples. This is not likely due to methodological differences between studies, as the modern calculus sample we analysed in this study (European descent) had a similar percentage of its sequences without alignment (19.4%) compared to ~25% for the (*Lloyd-Price et al., 2017*) study. One hypothesis for this finding is that modern reference databases are missing many oral microorganisms that were present in historical and ancient humans. Additionally, given that most modern microbiome studies and microbial genomes assembled are from European/American individuals (*Consortium, 2012*;

*Lloyd-Price et al., 2017*), current reference databases are likely missing oral microbial diversity from non-Industrial, non-Caucasian, or ancient human populations. Another possibility is that DNA contamination of dental calculus samples is from ancient or modern soil microorganisms that do not currently have reference sequences. Regardless of the cause, additional steps could be taken to improve the number of ancient DNA sequences that can be taxonomically identified. For example, de novo assembled genomes from these ancient samples could be used as reference sequences for further alignment-based taxonomic classification. Such tools currently exist (*Imelfort et al., 2014*), but their performance on short and degraded ancient DNA is yet to be determined. An alternative and complementary approach is to obtain a greater diversity of high-quality reference genomes from modern samples, including from non-Caucasian individuals. Until we can comfortably assign a higher proportion of ancient DNA sequences taxonomically, we recommend that palaeomicrobiological researchers report the percentage of unassigned sequences when classifying taxonomy and are aware of the fact that missing references can increase the rate of misclassifications (*Warinner et al., 2017*; *Velsko et al., 2018*).

Database sizes are a limitation for the currently implemented algorithms in MALT, as MALT uses large amounts of memory (e.g. >1 TB of RAM) when aligning sequences to the 2017nt and RefSeqGCS databases, and these requirements will increase as more genomes are added to databases. We were not able to investigate eukaryotic or viral classification in ancient metagenomes due to memory constraints, and instead focused on prokaryotes, which account for >99% of DNA in ancient dental calculus (*Warinner, Speller & Collins, 2014*; *Weyrich et al., 2017*). A possible solution may be better database curation, for example, through deduplication of the same strain with multiple entries, which could be accomplished using a sequence similarity clustering-based approach. Additionally, future algorithmic refinements in database compression may alleviate this issue. Ultimately, database choice is an essential facet of alignment-based taxonomic classification, and we urge researchers to carefully consider the pros and cons of different databases and how they can affect their findings. Additionally, database utilisation is a fluid issue; as more reference sequences are generated, reanalysis of palaeomicrobiological datasets will be important to reassess past interpretations and findings.

## CONCLUSIONS

Using both simulated and real data, this study demonstrated that nucleotide-to-protein alignments currently struggle to assign taxonomy to the short DNA fragments typical of ancient DNA. We found that nucleotide-to-nucleotide alignments using MALTn can faithfully recapitulate simulated metagenomes with high accuracy, even when reads are extremely short (30 bp) and contain high levels of deamination and random sequence divergence corresponding to 100,000 years of evolution. We also tested four different reference databases and find that database choice is an important factor to consider for alignment-based taxonomic classification in ancient metagenomic studies and that the application of full microbial references genomes within nucleotide alignment strategies currently produces the most robust results. Finally, we performed an in-depth

reanalysis of previously published paleomicrobiome studies, increasing the number of reads assigned taxonomy by an average of 64.2-fold and identifying taxa previously unidentified in the original study. We hope that the findings and suggestions provided in this paper will help inform future palaeomicrobiological researchers.

### Funding
This work was supported by the Australian Research Council (ARC): DECRA (DE150101574) and ARC Centre of Excellence CABAH (CE170100015). The funders had no role in study design, data collection and analysis, decision to publish, or preparation of the manuscript.

### Grant Disclosures
The following grant information was disclosed by the authors:
Australian Research Council (ARC): DECRA: DE150101574.
ARC Centre of Excellence CABAH: CE170100015.

### Competing Interests
The authors declare that they have no competing interests.

### Author Contributions

- Raphael Eisenhofer conceived and designed the experiments, performed the experiments, analysed the data, prepared figures and/or tables, authored or reviewed drafts of the paper, approved the final draft.
- Laura Susan Weyrich conceived and designed the experiments, contributed reagents/materials/analysis tools, authored or reviewed drafts of the paper, approved the final draft.

### Data Availability
Figshare:

Eisenhofer Philipona, Raphael (2018): List of assembly accessions for simulated MALT analyses. figshare. Dataset.

https://doi.org/10.25909/5b84c9c196f54.

Eisenhofer Philipona, Raphael (2018): Genomes-Used-For-Simulated-Metagenomes. figshare. Dataset.

https://doi.org/10.4225/55/5b0ca9b2cd6dc.

Eisenhofer Philipona, Raphael (2018): Simulated_Metagenomes. figshare. Dataset.
https://doi.org/10.4225/55/5b0caf73b7247.

Eisenhofer Philipona, Raphael (2018): RefSeqGCS genome accessions. figshare. Dataset.
https://doi.org/10.25909/5b84ddf58ac49.

Eisenhofer Philipona, Raphael (2018): HOMD accessions. figshare. Dataset.
https://doi.org/10.25909/5b84d19aaff2a.

## Supplemental Information

Supplemental information for this article can be found online at http://dx.doi.org/10.7717/peerj.6594#supplemental-information.

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
