# Peer review of "Assessing alignment-based taxonomic classification of ancient microbial DNA"

_PeerJ, doi:10.7717/peerj.6594_

## Round 0.1 · original submission · Major Revisions

Please address all the points raised by the two reviewers, including comparing misclassification rates to these provided in Velkso et al. 2018; addressing whether removing all species found in blanks is preferable to not doing so; accounting for possible caveats pertaining to uncultured organisms, and; downtone any interpretation of tree relationship in te absence of strong statistical support. Please also note that some conclusions are redundant with a recently published but uncited paper (Velkso et al. 2018). This article should be cited and discussed.

Reviewer 1 ·

Basic reporting

In their article "Assessing alignment-based taxonomic classification of ancient microbial DNA" the authors provide a comprehensive assessment of taxonomic characterisation of ancient DNA samples. They compare different methods, databases and quantify the effects of different properties of the data, such as DNA damage, variation in divergence etc. All simulated datasets are provided through figshare.
This study provides a valuable resource for the field.

Experimental design

no comment

Validity of the findings

no comment

Additional comments

I am wondering if the authors are aware of the study by Velsko et al. (Selection of Appropriate Metagenome Taxonomic Classifiers for Ancient Microbiome Research). I think this paper should be cited and the results should be put in relation with theirs.

In line 75 the authors state "Methods for analyzing shotgun sequencing data broadly fall into two categories: assembly-based and alignment-based."
I suggest that alignment-free methods are also mentioned here.

For running MALT the authors describe that they used blast2rma to convert MALT's blast output to rma.
As MALT is also producing rma output directly I am wondering why it was done this way.

About MALTx the authors say that it "is one of the few methods that can be used to assess microbial protein functionality in ancient metagenomic data sets."
However, in combination with functional annotations this is also possible when aligning to DNA databases. MEGAN, for example, has this functionality.

About the effect of divergence the manuscript states "However, this is at the theoretical limit of DNA preservation (Allentoft et al., 2012) and is thus unlikely to hamper most paleomicrobiological studies."
While this is true I believe the problem is not so much the divergence created by the age of the samples, but rather the database bias. I.e. the exact species is maybe not part of the database and the next relatives might have quite some divergence, which can lead to misassignments.

I have few more minor comments:

In line 36 the citation "Consortium, 2012" needs to be corrected.

"de novo" should be spelled without a hyphen (line 475).

There is a period in line 503 that needs to be removed.

Reviewer 2 ·

Basic reporting

This study is complementary to another recently published article (Velsko et al. 2018), and it comes to both similar (damage has little impact on taxonomic assignment) and different (the classification accuracy of MALTn) conclusions. However, this article is not cited or discussed. This paper would benefit from engaging with that study more directly and specifically explaining why some of the findings are different.

Velsko IM, Frantz LAF, Herbig A, Larson G, Warinner C. 2018. Selection of appropriate metagenome taxonomic classifiers for ancient microbiome research. mSystems 3:e00080-18. DOI: 10.1128/mSystems.00080-18.

Additionally:
Line 59: It is relevant to cite the recent Velkso et al. 2018 paper here, which also assesses taxonomic classification of simulated and empirical ancient metagenomes.

Line 89: Update Herbig et al. 2016 bioRxiv citation with Vagene et al. 2018. The latter is the published citation for MALT: Vågene AJ, Campana MG, Robles García N, Warinner C, Spyrou MA, Andrades Valtueña A, Huson D, Tuross N, Herbig A, Bos KI, Krause J. 2018. Salmonella enterica genomes recovered from victims of a major 16th century epidemic in Mexico. Nature Ecology and Evolution 1-9. DOI: 10.1038/s41559-017-0446-6.

Lines 104-107: The two findings listed here (deamination minimally impacts taxonomic classification and reference database choice is important) were also the conclusions of another recent paper (Velsko et al. 2018), which should be cited here. Another recent paper (Warinner et al. 2017) also reviews issues regarding alignment-based classification and database choice, and would be relevant to cite here: Warinner C, Herbig A, Mann AE, Fellows Yates JA, Weiss CL, Burbano HA, Orlando L, Krause J. 2017. A robust framework for microbial archaeology. Annual Reviews in Genomics and Human Genetics 18:321-356. http://www.annualreviews.org/doi/abs/10.1146/annurev-genom-091416-035526

Lines 253-255: Note that your results here are in agreement with previous findings (Velsko et al. 2018).

Experimental design

Using simulated metagenomes, this paper evaluates the performance of the DNA sequence aligner MALT, when run as MALTx and MALTn (genome and CDS), for microbial taxonomic classification. The influence of DNA damage and genomic evolution on taxonomic classification is also assessed. Finally, taxonomic assignment rates are characterized for a set of empirical ancient dental calculus metagenomes using four different databases (2014nr, 2017nt, HOMD, RefSeqGCS). This study concludes that using MALTn with the RefSeqGCS database yields the best results, and that applying this approach to previously published datasets alters the findings of some previous studies.

Overall, this study is a valuable and useful contribution to the field. The direct comparison of MALTx and MALTn is particularly useful. The comparison of databases is important, but the study would be greatly strengthened by performing this analysis on the simulated datasets as well, not just on experimental data.

Additional comments:

Line180-181: Why did you set the minimum support percent filter to 0.1% for ancient datasets and 0.01% for simulated metagenomes? Why not set both to 0.1%?

Line 186: Removing all species found in blanks seems ill-advised to me. First, if cross-contamination occurs at the bench, then oral bacteria are most likely to be contaminated into the blanks. Removal of these taxa from the whole dataset will then remove true oral bacteria from your samples. Additionally, if blanks were pooled with samples on the same Illumina lane (especially an Illumina 4000), barcode hopping during clustering could result in low level misclassification of sample reads as blank reads. If all taxa found in blanks are removed, this would then remove true sample IDs. For example, a given taxon may be present in the blanks with 35 reads (very few) but present in a sample with 35,000 reads (many). Given your current filtering scheme, would you really remove the 35,000 reads from your sample just because a few dozen reads were found in a blank? If that taxon were an important marker species, such as Fusobacterium nucleatum or Methanobrevibacter oralis, how would that affect your downstream analyses? Would it lead you to call a species absent when it is indeed present (but also present in blanks)? Please address whether the taxa you identify in your blanks include well-characterized oral bacteria and in which proportions.

At multiple places in the text (e.g., line 267, Figure 4, Table 1), you refer to the % of sequences being taxonomically assigned or not assigned. Clarify if you mean assigned in general or correctly assigned. Also, when “not assigned” do you mean not assigned to a species node or not assigned at all?

Line 277: In the section on how reference database affects taxonomic classification, only ancient datasets are tested. However, this should really be tested on your simulated dataset(s) in order to properly evaluate the results. Please test one or more of your simulated datasets against these four datasets. Otherwise, it is difficult to interpret the results of this section.

Validity of the findings

The misclassification rates you show for MALTn are much lower than recently observed in another similar paper comparing taxonomic classifiers using both simulated and empirical data (Velkso et al. 2018). Can you explain why your misclassification rate is so much lower? Is it related to the composition of your simulated metagenome? Or perhaps the database you are using? Or perhaps specific search parameters?

Lines 314-317 and 441-444: Note a caveat about uncultured organisms. The disadvantage of RefSeqGCS is that uncultured organisms, such as Saccharibacteria (TM7), are dramatically underrepresented, and these organisms include important oral taxa.

Lines 333-336: I like the inclusion of the tree and the reanalysis using MALTn with the RefSeqGCS database. However, I don’t think this statement about subsistence clustering is at all well supported by the tree. The tree shows little differentiation between hunter-gatherers and forager-gatherers (what is the difference between these names?), and additionally there is an LBK individual among several x-gatherers and a Neanderthal on the branch with agriculturalists. And why is LBK2 colored purple for hunter-gatherers? Additionally, modern humans cluster more closely to chimpanzees and a Neanderthal than to any ancient agriculturalist. Overall, there appears to be little structure in the tree.

Lines 343-346: As a caveat, I don’t think you test both deamination and evolution together. Perhaps rephrase?

Additional comments

Line 52: Age-associated deamination alters cytosine to thymine. Observed guanine to adenine substitutions are artifacts of PCR, created when a complementary strand is produced off of a deaminated template. Rephrase to clarify.

Line 104: For clarity, change “…taxonomic identification and…” to “…taxonomic identification compared to MALTx and…”

Lines 366-367: The meaning of this statement is unclear to me. Functional prediction can be made based on either DNA or amino acid sequences.

Line 395: It is true that UDG and partial UDG result in sequence shortening, but partial UDG does retain a damage signal. Perhaps rephrase?

---

## Round 0.2 · accepted · Accept

Dear Co-authors

The two reviewers and myself are satisfied with the revisions. I congratulate you for this much appreciated contribution to develop robust microbial profiling procedures in ancient DNA research. I have no doubt that the study will be of great benefit for the field.

Best
L

# Reviewer 1 ·

Basic reporting

The authors have submitted a new version of their paper "Assessing alignment-based taxonomic classification of ancient microbial DNA". The manuscript has improved substantially. All comments have been sufficiently addressed.

Experimental design

no comment

Validity of the findings

no comment

Reviewer 2 ·

Basic reporting

I am satisfied with the changes the authors have made in response to the reviews.

Experimental design

No comment

Validity of the findings

No comment

Additional comments

I have noticed only two minor things that require correction (but do not require re-review):

Line 95: Sentence needs rephrasing to provide a subject that is not in parentheses
Line 347 and 521: Italics should be removed for "sp. MM1"